# CHAMELEON: A Flexible Data-mixing Framework for Language Model Pretraining and Finetuning

Wanyun Xie [1]  Francesco Tonin [1]  Volkan Cevher [1]

## Abstract

Training data mixtures greatly impact the generalization performance of large language models. Existing domain reweighting methods often rely on costly weight computations and require retraining when new data is introduced. To this end, we introduce a flexible and efficient data mixing framework, CHAMELEON, that employs leverage scores to quantify domain importance within a learned embedding space. We first construct a domain affinity matrix over domain embeddings. The induced leverage scores determine a mixture that upweights domains sharing common representations in embedding space. This formulation allows direct transfer to new data by computing the new domain embeddings. In experiments, we demonstrate improvements over three key scenarios: (i) our computed weights improve performance on pretraining domains with a fraction of the compute of existing methods; (ii) CHAMELEON can adapt to data changes without proxy retraining, boosting few-shot reasoning accuracies when transferred to new data; (iii) our method enables efficient domain reweighting in finetuning, consistently improving test perplexity on all finetuning domains over uniform mixture. Our code is available at https://github.com/LIONS-EPFL/Chameleon.

## 1. Introduction

Pretraining large language models (LLMs) relies heavily on vast and diverse datasets, encompassing sources such as academic papers, books, and code repositories (Brown et al., 2020; Dubey et al., 2024). The composition of these datasets significantly influences the generalization capabilities and downstream performance of LLMs.

Domain reweighting, which involves adjusting the relative contributions of different domains in the training dataset, has emerged as a critical aspect of LLM training (Gao et al., 2020; Du et al., 2022). However, obtaining optimal domain weights is a challenging problem due to factors such as data quality, diversity, inter-domain overlap, and task-specific complexities (Shen et al., 2023; Longpre et al., 2024).

Early domain reweighting methods relied on manual selection, often favoring high-quality sources like Wikipedia and academic texts (Brown et al., 2020; Gao et al., 2020). While intuitive, these approaches are neither optimal nor scalable.

Recent work explores computational strategies for optimizing domain mixtures, such as DoReMi (Xie et al., 2023) and DoGE (Fan et al., 2024b), which use a small proxy model to derive domain weights for training a larger base model. Though effective, these methods are computationally expensive and have limited practical applicability.

We contend that an ideal data-mixing method should $(i)$ improve universal generalization, the fundamental goal of domain reweighting; $(ii)$ adapt to domain modifications – data naturally evolves between preparation and LLM training, making frequent recalibration impractical; $(iii)$ handle different training stages such as pertaining and fine-tuning. Most existing methods are limited to pretraining scenarios and do not consider either domain changes or the fine-tuning stage where domain specificity often plays a larger role.

In this work, we introduce CHAMELEON, a novel and efficient framework for data mixing that addresses these challenges. Our method computes domain weights directly from learned embeddings using *kernel ridge leverage scores* (KRLS), which quantify the importance of each domain based on its contribution to the overall embedding space.

Unlike existing approaches, CHAMELEON reduces the data-mixing compute and eliminates the need for frequent retraining of proxy models when domains change. Instead, it constructs a domain affinity matrix to capture relationships between domains and computes leverage scores that guide domain reweighting. Our data-centric approach enables

[1]Laboratory for Information and Inference Systems, École Polytechnique Fédérale de Lausanne (EPFL), Switzerland. Correspondence to: Wanyun Xie <wanyun.xie@epfl.ch>.

*Proceedings of the 42nd International Conference on Machine Learning*, Vancouver, Canada. PMLR 267, 2025. Copyright 2025 by the author(s).

seamless adaptation to new data and flexibility across both pretraining and fine-tuning stages.

Specifically, our contributions are summarized as follows:

- We propose CHAMELEON, an efficient data-mixing framework that leverages KRLS to quantify domain representativeness from embedded data. Inverse KRLS-based domain weights emphasize general knowledge for pertaining. We empirically demonstrate that CHAMELEON matches DoReMi and DoGE in pretraining performance at a fraction of their cost.

- As a data-centric method, CHAMELEON can flexibly adapt to changes in domain composition without retraining proxy models, enhancing its practicality in real scenarios. It outperforms baselines at 1% of the retraining cost, even as domains double.

- We extend domain reweighting to fine-tuning, where KRLS-based weights emphasize domain-specific uniqueness. Empirical results show consistent perplexity improvements across all domains on both natural language and code datasets.

From a practical standpoint, CHAMELEON significantly lowers the computational burden associated with domain reweighting, making advanced LLM training pipelines more accessible to researchers with limited resources.

Indeed, our KRLS-based scores are computationally inexpensive, hyperparameter-robust, and converge quickly. This efficiency is particularly advantageous when incorporating new data, where our method can be applied directly without re-running the entire proxy pipeline. By bridging the gap between pretraining and fine-tuning, our method provides a unified and agile framework for efficient as well as effective data mixing across all stages of LM training.

## 2. Related Work

**Domain Reweighting.** Domain reweighting improves LLM pretraining by balancing data contributions from different sources. In this setting, online adaptation strategies require frequent recalibration and monitoring (Albalak et al., 2023; Jiang et al., 2024; Fan et al., 2024a).

Two closely related approaches are DoReMi (Xie et al., 2023) and DoGE (Fan et al., 2024b). DoReMi trains both a reference and a proxy model using Group DRO (Sagawa et al., 2020) to mitigate excess domain loss, while DoGE tracks domain-specific gradients during proxy training.

Both methods are inefficient: DoReMi depends on the reference model's quality and requires training two models, while DoGE incurs high gradient tracking costs. Their domain weights fluctuate significantly during training (Figure 3).

Table 1: **Data-mixing methods capabilities comparison.**

|                | DoReMi | DoGE | CHAMELEON |
|----------------|:------:|:----:|:---------:|
| Generalization |   ✓    |  ✓   |     ✓     |
| Downstream     |   ✓    |  ✓   |     ✓     |
| New data       |   ✗    |  ✗   |     ✓     |
| Finetuning     |   ✗    |  ✗   |     ✓     |

In this setting, our work develops an offline method that achieves uniformly strong performance across domains without relying on downstream task knowledge. Other offline methods, such as Data Mixing Laws (Ye et al., 2024) and RegMix that–in contrast to ours–requires access to the downstream tasks (Liu et al., 2024), use multiple proxy models to search for optimal data mixtures, revealing that domain weights transfer across model sizes.

Additionally, studies on data and model scaling laws provide further insights into domain weighting strategies (Kang et al., 2024; Ye et al., 2024). However, they critically rely on the scaling strategy and do not have an easy way of adapting to domain expansion.

**Kernel Ridge Leverage Scores (KRLS).** The notion of statistical leverage score (Gareth et al., 2013) is used in best-rank approximation to define an importance score for the rows in a matrix by their influence on the optimal low-rank approximation (Mahoney & Drineas, 2009), with approximation error guarantees for sampling (Li et al., 2013).

Ridge leverage scores are also proposed with additional regularization term (Cohen et al., 2015; 2017; Rudi et al., 2018). Leverage scores are extended to the kernel setting in (Bach, 2013), namely kernel ridge leverage scores.

KRLS sampling is extensively used to accelerate kernel methods (Alaoui & Mahoney, 2015; Musco & Musco, 2017; Rudi et al., 2017; 2018). Inverse KRLS have been related to Christoffel functions (Pauwels et al., 2018), which in machine learning are used for, e.g., landmark sampling (Fanuel et al., 2022), density estimation (Pauwels et al., 2018), and outlier detection (Lasserre & Pauwels, 2019; Beckermann et al., 2021; Ducharlet et al., 2024).

## 3. Data Mixing via CHAMELEON

**Setup.** We consider a dataset $\mathcal{D} = \{D_1, \ldots, D_k\}$ consisting of $k$ distinct domains, each represented by its metadata (e.g., source, topic). Our objective is to determine a domain weight vector $\alpha \in \Delta_k$, where $\Delta_k$ is the probability simplex, enhancing the performance of LMs. Following prior work (Xie et al., 2023; Fan et al., 2024b), we adopt a two-stage strategy: (1) training a small proxy model to infer domain weights and (2) training a large base model using the com-

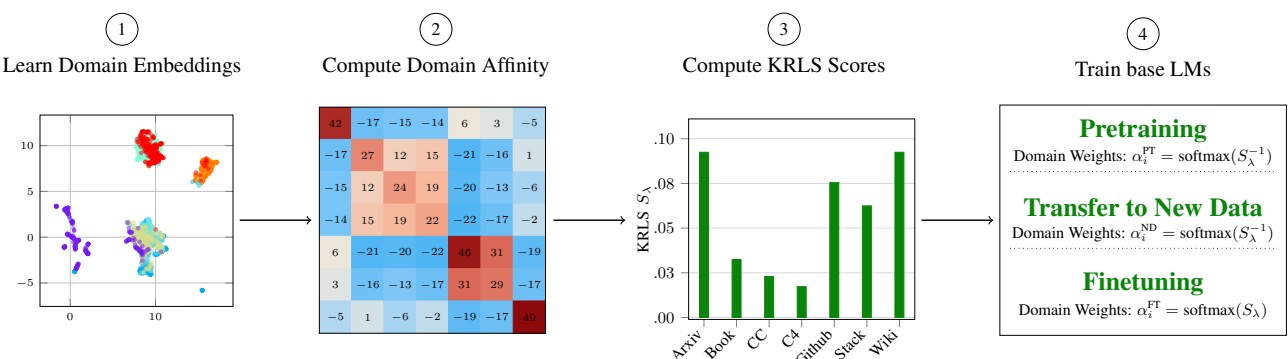

Figure 1: **Pipeline of domain reweighting via** CHAMELEON. The given data is first embedded through the proxy model, previously trained on a corpus $\mathcal{D}$ with uniform weights. Domain embeddings are then determined by averaging the embeddings for each domain. The domain affinity matrix $\Omega_{\mathcal{D}}$ is computed as the pairwise inner products between domain embeddings. Finally, (KRLS) is applied to $\Omega_{\mathcal{D}}$ to obtain score $S_{\lambda}$ indicating the degree of inter-domain independency. A resampling non-uniform distribution $\alpha$ is obtained by softmax normalizing the scores. Finally, the target base language model is trained with the obtained data mixture, where the inverse KRLS $S_{\lambda}^{-1}$ is used during pretraining of initial or new data to promote general knowledge learning and the KRLS $S_{\lambda}$ is used during finetuning to emphasize task-specific knowledge.

puted weights. Note that many studies have empirically shown that domain weights transfer across different model sizes (Xie et al., 2023; Fan et al., 2024b; Liu et al., 2024).

We approach the data mixture problem from a data-centric perspective. Unlike prior works such as DoReMi (Xie et al., 2023) and DoGE (Fan et al., 2024b), which derive domain weights based on the optimization process of a proxy model, we focus instead on the intrinsic properties of the data itself.

To characterize domain characteristics, we extract embeddings from hidden layers of the proxy model. These embeddings encapsulate rich semantic and structural information about the input data in a continuous, high-dimensional space. As a result, the embeddings not only represent the individual domains but also capture their inter-domain relationships.

Figure 1 ① presents a two-dimensional UMAP (McInnes et al., 2018) projection of mid-layer embeddings derived from the SlimPajama dataset using a proxy language model. The visualization highlights the following key characteristics: ($i$) semantic distinctiveness: similar domains cluster closely, while unique domains stand apart; ($ii$) centrality and coverage: broad domains like "CC" and "C4" create dominant regions, covering shared semantic space, while more specific domains like "Arxiv" are more distinct.

Our observations lead to two central questions:

1. *How can we precisely quantify domain characteristics?*

2. *How can such properties inform domain reweighting?*

We tackle these questions in the sequel.

### 3.1. Quantifying domain characteristics

We first introduce domain embeddings capturing domain-level characteristics, representing each domain $D_i$ as the embedding vector $x_i \in \mathbb{R}^p$, with $p$ being the embedding dimension. This embedding is computed by averaging the LM embeddings of data points in the domain:

$$x_i = \frac{1}{|D_i|} \sum_{a \in D_i} h_{\theta_p}^{(L)}(a),$$

where $h_{\theta_p}^{(L)}(a)$ denotes the $L$-th layer embedding of the proxy model $h_{\theta_p}$. In practice, using a sufficiently large, randomly sampled batch $B_i \subseteq D_i$ provides a robust approximation of the domain embedding and is used in experiments. The proxy is trained on $\mathcal{D}$ with uniform domain weights, i.e., $\alpha_i = 1/k$, following (Xie et al., 2023; Fan et al., 2024b). We define the resulting domain embedding matrix as $X = [x_1, \ldots, x_k]^\top \in \mathbb{R}^{k \times p}$.

To quantify properties across domains, we exploit Kernel Ridge Leverage Scores (KRLS). KRLS is a well-established tool in data analysis quantifying the influence or importance of data points (Alaoui & Mahoney, 2015). KRLS measure the contribution of each domain to the overall embedding space. First, we define a kernel function $\kappa(x_i, x_j) = x_i^\top x_j$, which captures the similarity between domains $D_i$ and $D_j$. Using this kernel, we construct the domain affinity matrix:

$$\Omega_{\mathcal{D}} = [\kappa(x_i, x_j)]_{i,j=1}^k = XX^\top.$$

The domain affinity matrix $\Omega_{\mathcal{D}}$ captures *pairwise* relationships between domains, with higher values indicating a higher degree of semantic similarity. Note that we

employ the linear kernel as the LM itself already introduces significant non-linearity. An example domain affinity matrix is visualized in Figure 1 ②. While $\Omega_{\mathcal{D}}$ captures inter-domain relationships, it does not directly provide a measure of *individual* domain importance for data mixing.

To address this, we propose to employ KRLS defined on $\Omega_{\mathcal{D}}$ to quantify the influence of each domain within the overall embedding space. We compute the scores as defined below.

**Definition 3.1** (Domain KRLS). For a given regularization parameter $\lambda > 0$, the KRLS for domain $D_i$ is defined as:

$$S_\lambda(D_i) = [\Omega_{\mathcal{D}}(\Omega_{\mathcal{D}} + k\lambda I)^{-1}]_{ii}, \quad \text{(KRLS)}$$

where $I$ is the $k \times k$ identity matrix.

The KRLS $S_\lambda(D_i)$ quantifies the importance of domain $D_i$ in the embedding space. Specifically, these scores correspond to the diagonal entries of the *hat* matrix $\Omega_{\mathcal{D}}(\Omega_{\mathcal{D}} + k\lambda I)^{-1}$ of Kernel Ridge Regression (KRR) (Hastie et al., 2005). They are proportional to the weights assigned to each domain in the dual KRR estimator, and they depend only on the inputs $x_i$ and constant $\lambda$ independently of specific target values (Calandriello et al., 2016; Chen & Yang, 2021). Inputs with higher KRLS indicate higher contribution to the KRR estimator, i.e., they are more unique in the kernel representation. More discussions on the KRLS and the role of regularization are provided in Appendix A.1 and A.2, respectively.

In data mixing, a high KRLS value for domain $D_i$ indicates that its embedding $x_i$ cannot be well-approximated as a combination of embeddings from other domains, implying that $D_i$ is relatively distinct or unique in its characteristics. Conversely, a low KRLS value suggests that $D_i$ is highly well-represented, as it can be readily reconstructed from other domain embeddings, representing broader or more widely shared characteristics across domains.

### 3.2. Incorporating KRLS into LM training

An essential question is thus: *Which domains should be prioritized: the general ones with higher degree of dependency with others or independent ones with more unique characteristics?* Prior work suggests that data mixing strategies should adapt to different training phases (Ma et al., 2023; Feng et al., 2024). Therefore, we address this by considering pretraining and fine-tuning separately, as their objectives differ fundamentally (Parthasarathy et al., 2024).

*Remark* 3.2 (Theoretical motivation). Before introducing our weights, we provide a theoretical motivation for our approach. It is well established that the inverse KRLS $S_\lambda^{-1}$ is proportional to the Christoffel function (Pauwels et al., 2018), which is a common measure of density of the data in embedding space. Christoffel functions precisely characterize the local density of the data distribution in the feature

---

**Algorithm 1** Domain Weighting via CHAMELEON.

---

1: **Input:** Training data from $k$ domains $\mathcal{D} = \{D_1, \ldots, D_k\}$, regularization parameter $\lambda$, embedding layer index $L$.
2: **if** Pretraining **then**
3:      Train proxy $h_{\theta_p}(a)$ with uniform weights $\alpha_i = \frac{1}{k}$.
4: **end if**
5: Extract domain embeddings: $x_i = \frac{1}{|D_i|} \sum_{a \in D_i} h_{\theta_p}^{(L)}(a)$ for each domain $D_i$.
6: Construct the feature matrix: $X = [x_1^\top, \ldots, x_k^\top]$.
7: Compute the domain affinity matrix: $\Omega_{\mathcal{D}} = XX^\top$.
8: Compute KRLS $S_\lambda(D_i)$ for each domain $D_i$ using $\Omega_{\mathcal{D}}$.
9: **if** Pretraining **then**
10:      Domain weights $\alpha_i^{\text{PT}} = \frac{\exp(S_\lambda^{-1}(D_i))}{\sum_{j=1}^k \exp(S_\lambda^{-1}(D_j))}$.
11: **else if** Fine-Tuning **then**
12:      Domain weights $\alpha_i^{\text{FT}} = \frac{\exp(S_\lambda(D_i))}{\sum_{j=1}^k \exp(S_\lambda(D_j))}$.
13: **end if**
14: **Output:** Domain weights $\alpha^{\text{PT}}$ or $\alpha^{\text{FT}}$.

---

space, where higher values indicate denser regions. Detailed remarks are provided in Appendix A.2.

**Pretraining.** During pretraining, assigning higher sampling probability to domains with high $S_\lambda^{-1}$ upweights high-density data regions, which are most influential on base LMs' performance (Mallen et al., 2023; Feng et al., 2024). To achieve this, we determine domain weights using the inverse of KRLS:

$$\alpha_i^{\text{PT}} = \frac{\exp\left(S_\lambda^{-1}(D_i)\right)}{\sum_{j=1}^k \exp\left(S_\lambda^{-1}(D_j)\right)},$$

where we convert the scores $S_\lambda^{-1}$ into the probability distribution $\alpha^{\text{PT}}$ by appliying softmax normalization $\frac{\exp(\cdot)}{\sum_{j=1}^k \exp(\cdot)}$.

Importantly, the domain weights obtained by CHAMELEON focus on the intrinsic properties of the data and our method does not affect the proxy model's training process. This allows to compute importance weights $\alpha_i^{\text{ND}}$ of **new domains** directly without requiring retraining of the proxy model by applying it to the new data. The proxy is used to obtain the new domain embeddings, from which the new (KRLS) score is calculated. In contrast, existing data mixture methods couple domain reweighting with the proxy model's optimization, necessitating costly retraining whenever domains are added. This dependency not only increases computational overhead but also contradicts the goal of improving efficiency in large-scale dynamic training pipelines.

**Finetuning.** Finetuning aims to specialize on a novel specific task (Yang et al., 2024), requiring the model to learn differential features not fully captured during pretraining,

so we instead prioritize the domains with high KRLS $S_\lambda$:

$$\alpha_i^{\text{FT}} = \frac{\exp\left(S_\lambda(D_i)\right)}{\sum_{j=1}^{k} \exp\left(S_\lambda(D_j)\right)}.$$

Note that the key difference between finetuning and pretraining in the data mixture problem is that, during finetuning, we do not need to train a separate proxy model. Instead, we directly use the pre-trained model for finetuning. This allows us to extract the embeddings from the pre-trained model for the relevant domains, which are then used to compute the domain weights.

**Algorithm and complexity analysis.** The overall domain reweighting process is summarized in Algorithm 1. Our phase-specific strategy ensures that it well adapts to the differing demands of pretraining and fine-tuning. Obtaining embeddings $x_i$ requires a single forward pass for each sample $a \in D_i$ through the proxy; inference is fast as the proxy is a small model. Given the typically small number of domains $k$, the KRLS computation in Definition 3.1 is cheap in $\mathcal{O}(k^3)$. We do not add any overhead in proxy training. Our approach is therefore efficient and contrasts with prior methods requiring domain-specific iterative optimization (Xie et al., 2023; Fan et al., 2024b).

## 4. Experiments

We show CHAMELEON improves the model's performance through data mixture with less computational costs (Section 4.1). In addition, CHAMELEON is scalable and can be applied to larger datasets without the need to retrain a proxy model (Section 4.2). Moreover, it can easily applied to fine-tuning tasks (Section 4.3).

### 4.1. Universal Generalization

Considering universal generalization, the main goal is to improve the general performance of the model across domains in the training set and also in various downstream tasks. For the performance on the in-distribution data, we measure the perplexity across all domains. For downstream tasks, we follow RegMix (Liu et al., 2024) selecting 13 tasks that cover various realistic scenarios.

**Training setup.** We experiment on the SlimPajama-627B dataset (Soboleva et al., 2023) consisting of 7 data domains. We choose Uniform with uniform domain weights, DoReMi, and DoGE as our baselines, which are downstream task agnostic offline methods same as CHAMELEON. We include RegMix as an additional reference, as it instead leverages prior knowledge of downstream tasks. Specifically, RegMix optimizes domain weights using the validation loss of the domain most correlated with downstream performance, identified as "CC" in their work (Liu et al., 2024).

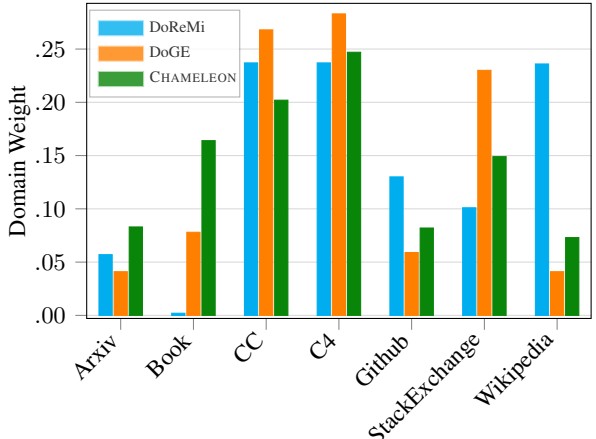

Figure 2: **Domain weights on SlimPajama**. We compare weights computed by data-mixing methods on SlimPajama.

Table 4: **GPU hours for obtaining domain weight.**

| DoReMi | DoGE | CHAMELEON | 684M base model |
|--------|------|-----------|-----------------|
| 7.4h | 6.3h | 0.8h | 56h |

Following DoGE (Fan et al., 2024b), we use a small 82M decoder-only transformer (Vaswani et al., 2017) as the auxiliary models for CHAMELEON, DoReMi, and DoGE. Auxiliary models for both DoGE and DoReMi are trained for 10k iterations, and the proxy model of CHAMELEON is only trained for 2k steps and we use 4k samples for embedding computation per domain. Detailed training setup is demonstrated in Appendix B.2. Domain weight obtained through different methods is reported in Figure 2. Through training small auxiliary models, domain weights are obtained to train larger base models with the size of 684M. Note that we employ the simple, linear kernel $\kappa(x_i, x_j) = x_i^\top x_j$, which does not need further kernel hyperparameter tuning.

**Perplexity and computational cost.** Table 2 shows per-domain perplexity on the held-out test dataset for 684M base models. CHAMELEON outperforms Uniform and DoReMi, achieving similar performance to DoGE but with significantly lower computational cost. Specifically, DoReMi trains two auxiliary models, while DoGE computes $k = 7$ gradients per iteration with roughly $1.7\times$ wall-clock time per iteration, both of which are costly. In contrast, CHAMELEON achieves competitive results with just 2k steps of training, and its inference cost is minimal compared to the training. In our setting, training an 82M proxy model requires $10^{17}$–$10^{18}$ FLOPs, and extracting embeddings requires only $10^{15}$ FLOPs ($< 1\%$ of proxy training). As a result, the total FLOPs of CHAMELEON, including both training and inference for embeddings, is only 10% of DoReMi

Table 2: **Universal generalization - perplexity.** Per-domain test perplexity for the universal generalization compared with the uniform baseline, DoReMi, DoGE, and RegMix with 684M parameter models. Note that, unlike other methods, RegMix knows the target downstream tasks for optimization. We report the compute (in FLOPs) required to arrive at the data mixture. CHAMELEON boosts generalization at a fraction of the computational cost.

| Domain | Uniform | DoReMi | DoGE | CHAMELEON | RegMix |
|---|---|---|---|---|---|
| Arxiv | 8.16 | 9.16 | 9.07 | 8.31 | 11.35 |
| Book | 42.55 | 46.48 | 40.30 | 39.23 | 41.52 |
| CC | 45.26 | 40.62 | 38.99 | 40.11 | 37.32 |
| C4 | 49.00 | 43.92 | 40.65 | 42.59 | 43.85 |
| Github | 3.99 | 4.10 | 4.09 | 4.20 | 4.99 |
| Stackexchange | 7.99 | 8.35 | 7.39 | 7.94 | 10.63 |
| Wikipedia | 12.42 | 10.78 | 15.74 | 13.90 | 20.88 |
| Average PPL ($\downarrow$) | 24.20 | 23.34 | 22.32 | **22.31** | 24.36 |
| # Domains Over Uniform | - | 3/7 | **4/7** | **4/7** | 3/7 |
| FLOPs | 0 | $1.34 \times 10^{18}$ | $6.68 \times 10^{17}$ | $\mathbf{1.36 \times 10^{17}}$ | $1.20 \times 10^{18}$ |
| | | $(10\times)$ | $(5\times)$ | $(1\times)$ | $(9\times)$ |

Table 3: **Universal generalization - reasoning.** Accuracy of downstream tasks in the same settings as Table 2.

| Benchmark | Unif. | DoReMi | DoGE | CH. | RegMix |
|---|---|---|---|---|---|
| ARC-E | 36.8 | 37.6 | 38.0 | 37.8 | 39.1 |
| COPA | 55.7 | 59.3 | 62.3 | 61.9 | 63.0 |
| HellaSwag | 26.5 | 27.0 | 27.2 | 27.1 | 27.0 |
| Lambada | 13.5 | 13.6 | 14.7 | 15.1 | 16.5 |
| LogiQA | 21.7 | 21.9 | 22.4 | 22.6 | 21.4 |
| MultiRC | 57.2 | 55.7 | 57.3 | 57.2 | 56.6 |
| OpenBook | 14.1 | 13.3 | 14.6 | 14.4 | 14.7 |
| PiQA | 59.2 | 59.5 | 60.0 | 60.5 | 57.6 |
| QQP | 36.8 | 36.8 | 36.8 | 39.2 | 37.1 |
| RACE | 26.1 | 25.3 | 26.4 | 26.5 | 27.3 |
| SciQ | 61.8 | 62.5 | 64.9 | 64.3 | 64.1 |
| Social IQA | 35.0 | 35.5 | 35.7 | 35.7 | 35.6 |
| WinoGrande | 50.5 | 51.3 | 52.0 | 52.1 | 50.9 |
| Average ($\uparrow$) | 37.9 | 38.4 | 39.4 | **39.6** | 39.3 |

Table 5: **Universal generalization with _1.2B_ model - reasoning.** Large-scale pretraining experiments.

| Benchmark | Unif. | DoReMi | DoGE | CH. | RegMix |
|---|---|---|---|---|---|
| ARC-E | 39.4 | 41.2 | 41.9 | 42.4 | 43.0 |
| COPA | 64.0 | 66.0 | 63.0 | 61.0 | 66.0 |
| HellaSwag | 27.5 | 27.7 | 28.2 | 28.4 | 27.6 |
| Lambada | 17.9 | 17.3 | 18.7 | 21.6 | 20.7 |
| LogiQA | 22.0 | 24.0 | 22.0 | 21.2 | 20.7 |
| MultiRC | 57.2 | 57.2 | 57.2 | 57.2 | 56.9 |
| OpenBookQA | 15.0 | 13.6 | 13.8 | 16.4 | 17.4 |
| PIQA | 61.5 | 61.9 | 61.8 | 63.8 | 58.7 |
| QQP | 36.8 | 36.8 | 36.9 | 36.9 | 36.8 |
| RACE | 26.0 | 26.7 | 27.8 | 29.1 | 28.4 |
| SciQ | 69.7 | 68.3 | 69.0 | 72.6 | 72.0 |
| SocialIQA | 36.2 | 36.5 | 35.9 | 37.2 | 36.1 |
| WinoGrande | 52.8 | 49.6 | 48.9 | 51.5 | 50.0 |
| Average ($\uparrow$) | 40.5 | 40.5 | 40.4 | **41.5** | 41.1 |

and 20% of DoGE. In addition, we demonstrate GPU hours in Table 4 and discuss its details in Appendix B.4, highlighting the efficiency and lower computational overhead of CHAMELEON.

**Evaluation on downstream tasks.** We apply our method on realistic downstream tasks. We follow RegMix (Liu et al., 2024) selecting 13 tasks that cover various realistic tasks: ARC-E (Clark et al., 2018), COPA (Sarlin et al., 2020), HellaSwag (Zellers et al., 2019), Lambada (Paperno et al., 2016), LogiQA (Liu et al., 2020), MultiRC (Khashabi et al., 2018), OpenBookQA (Mihaylov et al., 2018), PiQA (Bisk et al., 2020), QQP (Wang, 2018), RACE (Lai et al.,

2017), SciQ (Welbl et al., 2017), Social IQA (Sap et al., 2019), WinoGrande (Sakaguchi et al., 2021). The reported accuracy in Table 3 is the average from 0-shot to 5-shot evaluations following (Liu et al., 2024), scored using the lm-eval-harness evaluation framework (Gao et al., 2024).

These benchmarks cover a diverse range of tasks, enabling a comprehensive evaluation of the real-world impact of CHAMELEON. For each benchmark, we use normalized accuracy as the evaluation metric if provided by lm-eval-harness else we use regular accuracy. Notably, CHAMELEON also shows competitive performance across all downstream tasks even compared with RegMix, a task-aware method.

Table 6: **Universal generalization with *1.2B* model - perplexity.** Large-scale experiments in the same settings of Table 5.

| Domain | Uniform | DoReMi | DoGE | CHAMELEON | RegMix |
|---|---|---|---|---|---|
| Arxiv | 6.30 | 7.09 | 7.07 | 6.33 | 10.61 |
| Book | 28.25 | 32.66 | 27.83 | 24.63 | 27.55 |
| CC | 31.19 | 29.96 | 28.11 | 26.95 | 24.70 |
| C4 | 34.74 | 33.05 | 31.06 | 29.58 | 31.94 |
| Github | 2.91 | 3.03 | 3.07 | 2.94 | 4.08 |
| Stackexchange | 6.01 | 6.44 | 5.80 | 5.76 | 9.54 |
| Wikipedia | 8.65 | 7.93 | 10.88 | 9.03 | 20.08 |
| Average PPL (↓) | 16.86 | 17.17 | 16.26 | **15.03** | 18.36 |
| # Domains Over Uniform | - | 3/7 | **4/7** | **4/7** | 3/7 |

**Scale to 1.2B model.** Prior works (Xie et al., 2023; Fan et al., 2024b; Liu et al., 2024) have shown that domain weights transfer effectively across different model scales. To validate this for CHAMELEON, we extended our experiments to 1.2B models. The detailed results for perplexity and downstream task performance are presented in Table 6 and Table 5, respectively. Our findings confirm such transferability: the weights derived from an 82M proxy model proved effective when applied to both the 684M and 1.2B models. Importantly, we observed that CHAMELEON yielded even more significant improvements on larger models.

**Stability and Practicality.** We show that domain weights obtained by CHAMELEON remain stable across different training steps of the proxy model, whereas DoReMi and DoGE are sensitive or converge slowly as shown in Figure 3. Additionally, our method is robust to variations in proxy model size, the hyperparameter $\lambda$, and the number of samples computing embeddings, as detailed in Appendix B.5. This stability significantly reduces the need for extensive hyperparameter tuning, making CHAMELEON more practical and resource-efficient for real-world applications. In addition, we report GPU hours in Table 4 for domain weight computation and for training 684M base model, showing that the proxy training cost is non-negligible. Compared to DoReMi and DoGE, *we reduce computational overhead to less than 2% of final training cost*. This reduction is crucial for academic labs and smaller-scale training. More details are given in Appendix B.4.

### 4.2. Scalable to Pile

It is common for new data to be introduced during the official training of large base models, particularly when training on diverse and evolving datasets. However, existing methods including DoReMi, DoGE and RegMix require retraining a new proxy model from scratch whenever domains are added or removed, making them both inefficient and inconvenient for dynamic data environments. This process not only incurs significant computational costs but also delays the adaptation to changes in domain composition. *How can we develop a scalable method to reliably compute domain weights when domain composition changes?*

CHAMELEON has such scalability. Unlike DoReMi and DoGE, CHAMELEON's algorithm does not alter the proxy model's optimization, instead it focuses more on the intrinsic data characteristics, where the trained proxy model can already capture domain features, even for new unseen data.

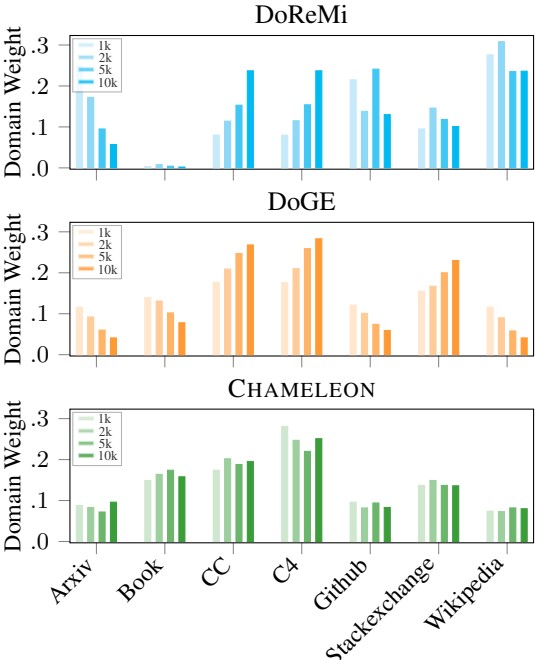

Figure 3: **KRLS scores converge quickly.** Domain weights across methods during proxy training. DoReMi and DoGE require more iterations to stabilize, while CHAMELEON converges quickly after 1k iterations. The detailed discussion is in Appendix B.5.

Table 7: **Transfer to new domains - perplexity.** Per-domain test perplexity when adapting to new data. The Human baseline is (Gao et al., 2020). When *domain composition changes*, the other methods need to retrain proxy models from scratch and re-run domain weight optimization, while we can directly compute the KRLS of the new domains by extending the proxy model trained on previous data. We report the extra compute (in FLOPs) required to adapt the data mixture to the Pile.

| Domain | Human | DoReMi | DoGE | CHAMELEON | RegMix |
|---|---|---|---|---|---|
| Arxiv | 9.76 | 14.11 | 10.78 | 9.73 | 16.19 |
| Dm_mathematics | 5.52 | 6.27 | 4.52 | 5.31 | 19.26 |
| Enron_emails | 12.82 | 9.96 | 9.39 | 7.77 | 12.06 |
| Europarl | 34.69 | 24.77 | 11.62 | 28.18 | 131.80 |
| Freelaw | 14.12 | 16.20 | 17.99 | 15.04 | 18.66 |
| Github | 5.92 | 5.84 | 4.90 | 6.16 | 9.95 |
| Gutenberg_pg_19 | 39.36 | 38.36 | 39.57 | 33.28 | 34.43 |
| Hackernews | 35.94 | 29.68 | 29.87 | 27.02 | 29.80 |
| Nih_exporter | 22.93 | 25.89 | 26.81 | 24.12 | 28.69 |
| Philpapers | 47.59 | 36.43 | 44.56 | 34.63 | 51.71 |
| Pile_cc | 43.19 | 32.85 | 58.17 | 34.90 | 32.30 |
| Pubmed_abstracts | 17.87 | 24.19 | 25.62 | 21.87 | 23.20 |
| Pubmed_central | 9.76 | 9.43 | 8.10 | 7.36 | 13.80 |
| Stackexchange | 10.41 | 11.48 | 11.79 | 11.25 | 18.96 |
| Ubuntu_irc | 36.12 | 32.10 | 23.20 | 29.34 | 20.71 |
| Uspto_backgrounds | 17.22 | 21.19 | 20.08 | 18.25 | 22.05 |
| Wikipedia_en | 28.70 | 25.95 | 40.42 | 24.68 | 29.32 |
| Average PPL ($\downarrow$) | 23.05 | 21.45 | 22.79 | **19.94** | 30.17 |
| # Domains Over Human | - | 9/17 | 10/17 | **11/17** | 4/17 |
| Extra FLOPs | 0 | $1.34 \times 10^{18}$ | $6.68 \times 10^{17}$ | $\mathbf{4.62 \times 10^{15}}$ | $3.5 \times 10^{18}$ |
| | | $(290\times)$ | $(145\times)$ | $(1\times)$ | $(758\times)$ |

Table 8: **Transfer to new domains - reasoning.** Accuracy of downstream tasks in the same settings as Table 7.

| Benchmark | Hum. | DoReMi | DoGE | CH. | RegMix |
|---|---|---|---|---|---|
| ARC-E | 37.5 | 39.3 | 35.3 | 39.2 | 39.5 |
| COPA | 56.8 | 61.5 | 54.7 | 60.9 | 61.2 |
| HellaSwag | 26.7 | 27.3 | 26.1 | 27.4 | 27.3 |
| Lambada | 12.5 | 15.8 | 9.3 | 15.9 | 15.4 |
| LogiQA | 22.5 | 21.2 | 22.1 | 23.8 | 21.9 |
| MultiRC | 56.7 | 56.5 | 57.2 | 57.3 | 56.2 |
| OpenBook | 13.3 | 13.1 | 13.1 | 14.2 | 14.5 |
| PiQA | 57.8 | 59.3 | 55.9 | 59.7 | 60.4 |
| QQP | 37.5 | 36.8 | 36.8 | 37.2 | 37.6 |
| RACE | 25.8 | 27.2 | 24.9 | 26.8 | 27.2 |
| SciQ | 64.1 | 65.7 | 58.1 | 66.0 | 67.1 |
| Social IQA | 35.0 | 36.0 | 34.2 | 36.6 | 36.3 |
| WinoGrande | 50.7 | 51.2 | 49.8 | 50.9 | 49.9 |
| Average ($\uparrow$) | 38.2 | 39.3 | 36.7 | **39.7** | 39.6 |

To test its scalability, we employ the proxy model trained on Slimpajama in Section 4.1 to Pile dataset (Gao et al., 2020) directly. Both SlimPajama and Pile are large-scale datasets used for pretraining LMs, with overlapping data sources such as books, scientific texts, web content, and codebases. The Pile dataset includes more domains than Slimpajama and its data is more diverse. Note that the original Pile dataset includes 22 domains but only 17 are now available due to copyright issues.

To obtain domain weights of Pile, we input 4k samples per domain of Pile to the proxy model trained on Slimpajama in Section 4.1 and infer embeddings for computing domain weights through (KRLS). The computed domain weights are reported in Table 19, where we use the domain weights reported in their respective papers for DoReMi and RegMix. We use Human as baseline that is selected manually as in the Pile paper (Gao et al., 2020). As in Section 4.1, we report per-domain perplexity in Table 7 and downstream accuracy in Table 8.

CHAMELEON outperforms DoReMi, DoGE and even Reg-Mix in both cases. Importantly, FLOPs of CHAMELEON is marginal compared with DoReMi and DoGE (only 1% of training a new proxy model from scratch) since we can reuse the previous proxy model and our extra FLOPs only include inference cost for extracting embeddings, while DoReMi DoGE, and RegMix require retraining the proxy models.

Table 9: **Finetuning.** Per-domain test perplexity compared with the uniform baseline for finetuning on the 7 languages in Wiki40b. CHAMELEON can flexibly be extended to the finetuning pipeline by computing the KRLS of the finetuning domains, improving performance across all domains.

| Domain | Uniform | CHAMELEON |
|---|---|---|
| French | 6.86 | 6.51 |
| German | 10.12 | 8.78 |
| Italian | 13.29 | 12.42 |
| Spanish | 8.41 | 8.04 |
| Portuguese | 8.00 | 7.78 |
| Dutch | 13.98 | 12.30 |
| Polish | 5.07 | 4.21 |
| Average PPL ($\downarrow$) | 9.43 | **8.58** |
| # Domains Over Uniform | - | 7/7 |

Table 10: **Finetuning.** Per-domain test PPL vs. the uniform baseline for finetuning on the 7 programming languages in Stack dataset. CHAMELEON improves across all domains.

| Domain | Uniform | CHAMELEON |
|---|---|---|
| Python | 19.98 | 16.53 |
| Java | 19.27 | 15.53 |
| C | 28.24 | 22.58 |
| C++ | 25.16 | 21.09 |
| Go | 30.25 | 19.26 |
| Ruby | 21.78 | 17.83 |
| PHP | 9.45 | 7.43 |
| Average PPL ($\downarrow$) | 22.02 | **17.18** |
| # Domains Over Uniform | - | 7/7 |

Note that as the number of domains increases, DoGE computes $k = 17$ gradients per iteration, resulting in approximately $2.5\times$ wall-clock time per iteration on this dataset.

### 4.3. Finetuning

Besides pertaining, CHAMELEON interestingly allows data-mixture optimization during finetuning. We take advantage of the existing pretrained model and can extract embeddings on finetuning data directly for domain weight computation. As discussed in Section 3.2, the goals of pretraining and fine-tuning are distinct, with pretraining aiming for broad generalization and fine-tuning focusing on specialization. Therefore, we directly use leverage scores for computing domain weights, as described in Section 3.2.

We fine-tune a pretrained model trained on the Pile (from Section 4.2) for 10k steps on two separate datasets: (*i*) Wiki40b (Guo et al., 2020), which includes multiple languages, for which we select 7 Latin languages, and (*ii*) Stack-decup (Kocetkov et al., 2022), which covers various programming languages, from which we use 7. The results are shown in Table 9 and Table 10. CHAMELEON outperforms the uniform weights baseline across all domains in both tasks, showing our data mixture can greatly benefit finetuning. Remarkably, our weight computation is computationally cheap in finetuning, as we simply need forward passes through the pretrained model to compute domain embeddings and we can then directly apply (KRLS).

Additionally, we present fine-tuning results using $\alpha^{PT}$ instead of $\alpha^{FT}$ in Appendix B.9 for reference. The results demonstrate that fine-tuning with KRLS-based domain weights outperforms using their inverse. This indicates that data-mixing strategies should be tailored to different training phases.

## 5. Conclusion

We introduce CHAMELEON, a novel and efficient framework for data mixing that leverages KRLS to quantify the representativeness of data domains. We demonstrate that inverse KRLS-based domain weights effectively identify highly important domains for pretraining LMs. CHAMELEON can adapt to new domains without retraining proxy models, outperforming baselines in downstream tasks. Given that it is computationally inexpensive and stable, CHAMELEON lowers the overall cost of the expensive LLM pretraining pipeline, which can be useful both in industry and within academic budgets. We also extend domain reweighting to fine-tuning with KRLS-based weights, demonstrating consistent improvements.

This work highlights the need to tailor data-mixing strategies to different training phases. In future work, we aim to extend our approach to online settings for dynamic optimization during training. Additionally, we will extend to target specific downstream tasks by modifying the identity matrix within the KRLS to emphasize relevant domains, enhancing our method's flexibility for specific downstream tasks.

## Acknowledgements

We thank the reviewers for their constructive feedback. Thanks to Simin Fan for the helpful discussion. This work was supported as part of the Swiss AI Initiative by a grant from the Swiss National Supercomputing Centre (CSCS) under project ID a06 on Alps. This work was supported by the Swiss National Science Foundation (SNSF) under grant number 200021_205011. This work was supported by Hasler Foundation Program: Hasler Responsible AI (project number 21043). Research was sponsored by the Army Research Office and was accomplished under Grant Number W911NF-24-1-0048.

## Impact Statement

The approach presented in this paper aims at advancing the field of Machine Learning. No other potential societal consequences of our work are deemed necessary to specifically highlight here.

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

# Appendix

## Table of Contents

# A. Additional discussions on the KRLS

## A.1. Details of leverage scores

In this work, we employ KRLS to assign scores to data domains, and we focus on the leverage scores as a measure of domain importance, both in pretraining and finetuning language models. The KRLS (Alaoui & Mahoney, 2015) is a kernelized-version of the ridge leverage scores, which are used to quantify the importance of the rows in a matrix for the best-rank approximation with approximation error guarantees (Mahoney & Drineas, 2009; Li et al., 2013). The KRLS for the $i$-th domain is defined as

$$S_i = \left(\Omega_{\mathcal{D}}(\Omega_{\mathcal{D}} + k\lambda I)^{-1}\right)_{ii}, \tag{1}$$

with kernel matrix $\Omega_{\mathcal{D}ij} = \kappa(x_i, x_j)$ and regularization constant $\lambda > 0$. In our data-centric approach, we rank domains based on the degree of inter-domain dependency. To better see this, we establish the equivalence of RLS computed in the feature space induced by the finite-dimensional feature map $\phi$ and KRLS when employing the corresponding kernel $\kappa(x_i, x_j) = \phi(x_i)^\top \phi(x_j)$, where we use $\phi(x) = x$ in Section 3.1. This result provides insights into the behavior of the employed KRLS for domain reweighting. Let the set of $k$ domain embeddings $\{x_i\}_{i=1}^k$, where $x_i \in \mathbb{R}^p$. Let $\phi : \mathbb{R}^p \to \mathbb{R}^d$ be a finite-dimensional feature map with associated p.s.d. kernel $\kappa(x_i, x_j) = \phi(x_i)^T \phi(x_j)$ by kernel trick (Vapnik, 1999), and $\Phi(X) \in \mathbb{R}^{k \times d}$ be the matrix whose rows are the feature mappings $\phi(x_i)^\top$. The ridge regression hat matrix in feature space is $H_\lambda^\phi = \Phi(X)(\Phi(X)^T \Phi(X) + \lambda I)^{-1}\Phi(X)^T$, with ridge leverage scores $\mathrm{diag}(H_\lambda^\phi)$. In kernel ridge regression, the kernel ridge hat matrix is $H_\kappa = \Omega_{\mathcal{D}}(\Omega_{\mathcal{D}} + \lambda I)^{-1}$ and the kernel ridge leverage scores are given by $\mathrm{diag}(H_\kappa)$.

**Lemma A.1.** *With kernel $\kappa(x, y) = \phi(x)^T \phi(y)$, where $\phi : \mathbb{R}^d \to \mathbb{R}^p$ is a finite-dimensional feature map, the RLS in feature space induced by $\phi$ and the KRLS are s.t. $\mathrm{diag}(H_\lambda^\phi) = \mathrm{diag}(H_\kappa)$.*

*Proof.* With the kernel $\kappa(x, y) = \phi(x)^T \phi(y)$, we have $\Omega_{\mathcal{D}} = \Phi(X)\Phi(X)^T$. Substituting into $H_\kappa$:

$$H_\kappa = (\Phi(X)\Phi(X)^T)(\Phi(X)\Phi(X)^T + \lambda I)^{-1}.$$

We utilize the following matrix identity: for matrices $A \in \mathbb{R}^{m \times n}$ and $B \in \mathbb{R}^{n \times m}$ and a scalar $\lambda \neq 0$,

$$A(BA + \lambda I)^{-1} = (AB + \lambda I)^{-1}A.$$

This identity can be verified by multiplying both sides by $(BA + \lambda I)$ from the right, then by $(AB + \lambda I)$ from the left, which yields the same result on both sides.

Let $A = \Phi(X)$ and $B = \Phi(X)^T$. Then:

$$\Phi(X)(\Phi(X)^T \Phi(X) + \lambda I)^{-1} = (\Phi(X)\Phi(X)^T + \lambda I)^{-1}\Phi(X).$$

Therefore,

$$H_\lambda^\phi = \Phi(X)(\Phi(X)^T \Phi(X) + \lambda I)^{-1}\Phi(X)^T = (\Phi(X)\Phi(X)^T + \lambda I)^{-1}\Phi(X)\Phi(X)^T = H_\kappa.$$

Thus, the diagonal elements, and hence the leverage scores, are equivalent: $\mathrm{diag}(H_\lambda^\phi) = \mathrm{diag}(H_\kappa)$. $\square$

The above lemma establishes the equivalence of the ridge leverage scores computed in the feature space induced by the finite-dimensional feature map $\phi$ and the KRLS when employing the corresponding kernel $\kappa(x_i, x_j) = \phi(x_i)^\top \phi(x_j)$. It is possible to relate the RLS to the ridgeless solution, where the regularized solution converges to the least-norm solution as $\lambda \to 0$. Let the eigendecomposition of $\Omega_{\mathcal{D}} = U\Sigma U^\top$, then the KRLS of domain $i$, $i = 1, \ldots, k$ can be written as

$$S_\lambda(D_i) = \sum_{j=1}^k \frac{\sigma_j}{\sigma_j + \lambda} U_{ij}^2,$$

where $\sigma_j$ is the $j$-th eigenvalue of $\Omega_{\mathcal{D}}$. Therefore, the KRLS is a weighted version of the standard statistical leverage (Gareth et al., 2013), i.e., $\sum_{j=1}^k U_{ij}^2$, with weights depending on the regularization and the eigenspectrum of $\Omega_{\mathcal{D}}$. We now recall the relationship between the least-norm solution of a system of equations to the (ridgeless) statistical leverage score $\ell_i = \phi(x_i)^\top (\Phi(X)^\top \Phi(X))^+ \phi(x_i)$ (Mahoney & Drineas, 2009).

**Lemma A.2.** *The least-norm solution characterizes $\ell_i$ as follows:*

$$\ell_i = \min_{\Phi(X)^\top y = \phi(x_i)} \|y\|_2^2. \tag{2}$$

The leverage score of domain $i$ measures how important it is in composing the row space of $\Phi(X)$. If a row (domain) has a component orthogonal to all other rows (domains), its leverage score is 1. In data mixture, (2) seeks a linear combination of features that best approximates the embedding $x_i$ of the $i$-th domain. Intuitively, $\ell_i$ is highest when $x_i$ is linearly independent from the other domain embeddings. In our approach, we use the KRLS to assign scores to data domains both in pretraining and finetuning language models. High scores indicate data domains that are difficult to approximate with a linear combination of other domains, and are thus more unique. On the other hand, low scores indicate data domains that show higher degree of dependency with other domains, identifying more common data characteristics. During pretraining, we rank domains with lower KRLS as more important, as they are more common and thus more useful for learning general-purpose language representations. During finetuning to a specific task, we upweight domains with higher KRLS, as they are more unique and therefore better suited for learning task-specific representations.

### A.2. Discussions

We now discuss additional theoretical insights and properties of our methodology.

- **Role of regularization.** Adding $\lambda I$ to $\Omega_{\mathcal{D}}$ in (1) reduces the influence of the small principal components, resulting in proportionally lower sampling probability. A large $\lambda$ soft-thresholds the low part of the spectrum of $\Omega_{\mathcal{D}}$ and amplifies the contribution of the top eigenvectors of the kernel matrix, focusing on the most dominant domains. In practice, due to the relatively small number of domains, we observe that our algorithm is robust to the choice of $\lambda$, where small regularization is sufficient and larger values do not significantly alter the computed domain weights.

- **Adaptability to larger models.** Our approach can robustly transfer domain weights obtained from a small proxy model $h_{\theta_p}$ to a larger target model $h_{\theta_t}$ thanks to its use of domain affinities. Specifically, our method computes domain weights based on the kernel matrix $\Omega_{\mathcal{D}}$, which captures pairwise inner products between domain embeddings. While the absolute embeddings $x_i$ may vary between the proxy and target model, their inner products show a higher degree of consistency across model sizes. For instance, if the proxy learns that the "github" and "stackexchange" domains are semantically close, a larger, more powerful model typically also maintains this proximity in embedding space. Consequently, the pairwise similarities encoded in $\Omega_{\mathcal{D}}$, and therefore the resulting KRLS scores (1) and domain weights, are robust across different model scales. Empirical evidence is presented in Table 16.

- **Relation with Christoffel functions.** In machine learning literature, the *Christoffel function* is a key concept that characterizes the local density of the data distribution in feature space (Pauwels et al., 2018). Christoffel functions are known in orthogonal polynomials (Dunkl & Xu, 2014) and approximation theory (De Marchi et al., 2014). They are extended to machine learning (Pauwels et al., 2018), where it makes the connection between inverse leverage scores and the kernelized Christoffel function. In machine learning, they are mainly used for landmark sampling (Fanuel et al., 2022), density estimation (Pauwels et al., 2018), and outlier detection (Lasserre & Pauwels, 2019; Beckermann et al., 2021; Ducharlet et al., 2024). Given samples $\{z_i\}_{j=1}^n$, the kernelized Christoffel function is defined as the following regularized minimization over a reproducing kernel Hilbert space (RKHS) $\mathcal{H}$ with associated kernel $\kappa_{\mathcal{H}}$:

$$C_{\lambda,\eta}(z) = \inf_{g \in \mathcal{H}} \sum_{j=1}^n \frac{\eta_j}{n} g(z_i)^2 + \lambda \|g\|_{\mathcal{H}}^2 \quad \text{s.t.} \quad g(z) = 1, \tag{3}$$

where $\lambda > 0$ is a regularization constant, $\|g\|_{\mathcal{H}}$ denotes the RKHS norm of $g$, and $\eta_j > 0$. The Christoffel function is linked to the ridge leverage scores (RLS) (Alaoui & Mahoney, 2015; Cohen et al., 2017; Rudi et al., 2018), which quantify the influence of each sample on the learned model. Specifically, the Christoffel function at a point is proportional to the inverse of its RLS. High RLS values indicate data points that are difficult to represent as linear combinations of other points in the feature space. Conversely, a high Christoffel function value (and thus a low RLS) suggests a data point lies in a region of high data density and can be better expressed in terms of other points. In our work, focusing on domains with high degree of *linear dependency*, i.e., high Christoffel function, is shown to enable improved generalization and transfer learning capabilities. The following lemma details the closed-form expression for the regularized kernelized Christoffel function at each sample.

**Lemma A.3** ((Pauwels et al., 2018)). *The regularized kernelized Christoffel function takes the following value at sample $j$, for $j = 1, \ldots, n$:*

$$C_{\lambda,\eta}(z_j) = \frac{\eta_j}{n} \left( K \left( K + n\lambda \, diag(\eta)^{-1} \right)^{-1} \right)_{jj}^{-1}, \tag{4}$$

*where $K \in \mathbb{R}^{n \times n}$ is the kernel matrix with entries $K_{ij} = \kappa_{\mathcal{H}}(z_i, z_j)$, and $diag(\eta) = diag(\eta_1, \ldots, \eta_n)$ is a diagonal matrix with entries $\eta_j$ on the diagonal.*

This is derived from the representer theorem applied to (3) (Schölkopf et al., 2001; Pauwels et al., 2018). Lemma A.3 reveals the connection between the Christoffel function and the KRLS (1). The Christoffel function $C_{\lambda,1}(z_j)$ is therefore inversely proportional to KRLS of sample $z_j$ with $\eta = 1$.

## B. Additional experiments

### B.1. Additional figures for Figure 1

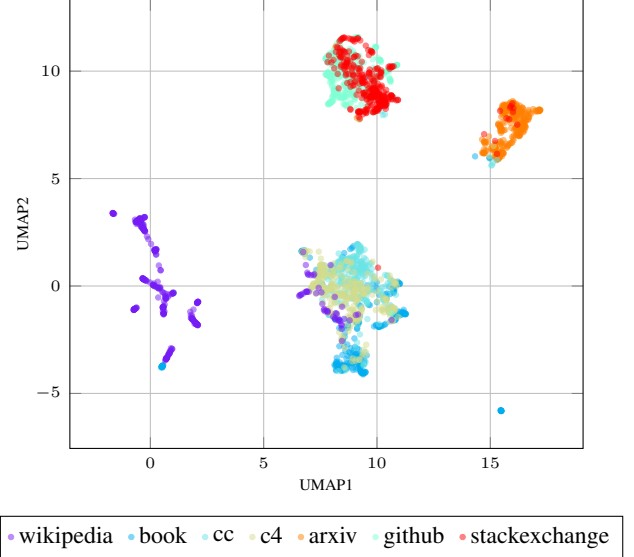

Figure 4: **Domains in embedding space**. 2D UMAP visualization of embeddings of SlimPajama learned by the proxy model. Semantically similar domains occupy similar regions in embedding space, creating high-density clusters.

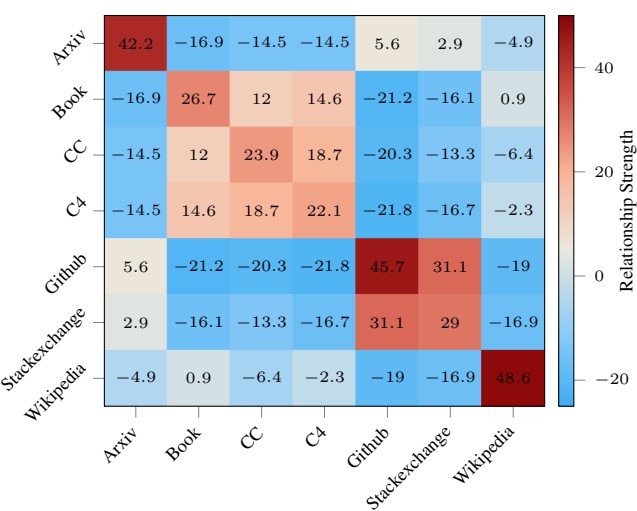

Figure 5: **Domain affinity matrix**. The matrix shows the relationship strength between domains in SlimPajama.

### B.2. Experimental setup.

We follow the experimental setup of DoGE (Fan et al., 2024b), using a small 82M decoder-only Transformer (Vaswani et al., 2017) as the auxiliary model for CHAMELEON, DoReMi, and DoGE. Additionally, we use a 684M model as the base model for pretraining. Moreover, we set the batch size 128, the cosine learning rate scheduler, weight decay 0.01, and gradient clipping 1.0 for all models. For the training on Slimpajama, Wiki40b, and Stack datasets, we set batch size 128. We increase the batch size to 512 on the Pile dataset since it is more noisy and has a larger number of domains.

In CHAMELEON, a temperature factor $\tau$ in the softmax normalization for domain weights is additionally applicable: $\alpha_i = \exp(z_i/\tau) / \sum_{j=1}^{k} \exp(z_j/\tau)$, where $z = S_{\lambda}^{-1}(D)$ for pretraining and $z = S_{\lambda}(D)$ for fine-tuning. In our experiments, we typically set $\tau^{\text{PT}} \in [5, 10]$ and $\tau^{\text{FT}} \in [0.2, 0.5]$.

For the ablation study of CHAMELEON, we also evaluate proxy models with other sizes (60M, 124M, and 210M). The model architectures and their corresponding learning rates are reported in Table 11.

For the setup of RegMix on the Slimpajama dataset, we follow its original setup (Liu et al., 2024) and train 200 1M proxy models with 1k steps to fit a regression model.

Table 11: **Architecture hyperparameters for various model scales.**

|       | Layers | Attention heads | Embed dim | Hidden dim | Max. learning rate (min.) |
|-------|--------|-----------------|-----------|------------|---------------------------|
| 60M   | 3      | 6               | 768       | 3072       | $5 \times 10^{-4}$ ( $1 \times 10^{-4}$ ) |
| 82M   | 6      | 12              | 768       | 3072       | $5 \times 10^{-4}$ ( $1 \times 10^{-4}$ ) |
| 124M  | 12     | 12              | 768       | 3072       | $5 \times 10^{-4}$ ( $1 \times 10^{-4}$ ) |
| 210M  | 24     | 16              | 768       | 3072       | $5 \times 10^{-4}$ ( $1 \times 10^{-4}$ ) |
| 684M  | 36     | 24              | 1200      | 4800       | $1.5 \times 10^{-4}$ ( $5 \times 10^{-5}$ ) |
| 1.2B  | 36     | 25              | 1600      | 6400       | $1.5 \times 10^{-4}$ ( $5 \times 10^{-5}$ ) |

### B.3. Domain weights on Slimpajama

We report our final domain weights for base model training in Table 12. Specifically, DoReMi and DoGE use domain weights through training proxy models with 10k steps. CHAMELEON use the model with 2k steps. For RegMix, we follow its paper (Liu et al., 2024) using "CC" as the target domain and train 200 1M proxy models to get the domain weights.

Table 12: **Final domain weights.**

| Domain | DoReMi | DoGE | CHAMELEON | RegMix |
|--------|--------|------|-----------|--------|
| Arxiv         | 0.057 | 0.041 | 0.083 | 0.001 |
| Book          | 0.002 | 0.078 | 0.164 | 0.025 |
| CC            | 0.237 | 0.268 | 0.202 | 0.924 |
| C4            | 0.237 | 0.283 | 0.247 | 0.024 |
| Github        | 0.130 | 0.059 | 0.082 | 0.019 |
| Stackexchange | 0.101 | 0.230 | 0.149 | 0.006 |
| Wikipedia     | 0.236 | 0.041 | 0.073 | 0.001 |

### B.4. Temporal cost

In Table 13, we report the required GPU (H100) hours for obtaining domain weight regarding the experiments in Table 2. We also show GPU hours of training 684M base model for a reference. We claim that improving the efficiency of determining the domain mixture is essential because the associated computational cost of proxy training is non-negligible. Compared to DoReMi and DoGE, which add over 10% to base model training costs, *we reduce computational overhead to less than 2% of final training cost*. This reduction is crucial for academic labs and smaller-scale training.

More importantly, the computational cost reported is often an optimistic lower bound for the baselines even for larger base models, since *DoReMi and DoGE require extensive hyperparameter tuning*. It has been shown that DoReMi's weights are unstable or difficult to reproduce (Fan et al., 2024b; Parmar et al., 2024) and DoGE approximations make it more sensitive to learning rate (Kang et al., 2024). Such sensitivity necessitates repeated validation on base models. Nevertheless, CHAMELEON is insensitive to hyperparameters and a detailed discussion is in Appendix B.5.

Table 13: **GPU hours for obtaining domain weight.**

| DoReMi | DoGE | CHAMELEON | 684M base model |
|--------|------|-----------|-----------------|
| 7.4h   | 6.3h | 0.8h      | 56h             |

### B.5. Stable domain weights of CHAMELEON

Table 14 corresponds to Figure 3 and reports the specific domain weights obtained by the proxy model with the different **number of steps**. CHAMELEON is more stable and converges faster than DoReMi and DoGE. In addition, we report perplexity using domain weights derived from a proxy model trained for 2k steps for DoReMi and DoGE in Table 15. This further demonstrates that DoReMi and DoGE converge slower than CHAMELEON.

Furthermore, Table 16 demonstrates that CHAMELEON is also very stable across different **model sizes**, $\lambda$ **values**, and **number of samples** for embedding computations.

Table 14: **Domain weights at different steps.**

| Domain | DoReMi | | | | DOGE | | | | CHAMELEON | | | |
|---|---|---|---|---|---|---|---|---|---|---|---|---|
| | 1k | 2k | 5k | 10k | 1k | 2k | 5k | 10k | 1k | 2k | 5k | 10k |
| Arxiv | 0.251 | 0.172 | 0.095 | 0.057 | 0.116 | 0.092 | 0.060 | 0.041 | 0.088 | 0.083 | 0.072 | 0.096 |
| Book | 0.003 | 0.008 | 0.004 | 0.002 | 0.139 | 0.131 | 0.102 | 0.078 | 0.149 | 0.164 | 0.174 | 0.158 |
| CC | 0.080 | 0.114 | 0.153 | 0.237 | 0.177 | 0.209 | 0.247 | 0.268 | 0.174 | 0.202 | 0.188 | 0.195 |
| C4 | 0.080 | 0.115 | 0.154 | 0.237 | 0.176 | 0.210 | 0.259 | 0.283 | 0.281 | 0.247 | 0.220 | 0.251 |
| Github | 0.215 | 0.138 | 0.241 | 0.130 | 0.121 | 0.101 | 0.074 | 0.059 | 0.096 | 0.082 | 0.094 | 0.083 |
| Stackexchange | 0.095 | 0.146 | 0.118 | 0.101 | 0.155 | 0.167 | 0.200 | 0.230 | 0.137 | 0.149 | 0.137 | 0.136 |
| Wikipedia | 0.276 | 0.308 | 0.235 | 0.236 | 0.116 | 0.090 | 0.058 | 0.041 | 0.074 | 0.073 | 0.082 | 0.080 |

Table 15: **Perplexity using domain weights derived from a proxy model trained for 2k steps.**

| Domain | DoReMi | DoGE | CHAMELEON |
|---|---|---|---|
| Arxiv | 8.08 | 8.49 | 8.31 |
| Book | 52.07 | 40.38 | 39.23 |
| CC | 48.69 | 41.31 | 40.11 |
| C4 | 52.98 | 44.54 | 42.59 |
| Github | 3.99 | 4.05 | 4.20 |
| Stackexchange | 7.98 | 7.81 | 7.94 |
| Wikipedia | 10.57 | 13.98 | 13.90 |
| Average PPL ($\downarrow$) | 26.34 | 23.01 | 22.31 |

Table 16: **Domain weights across different model sizes, $\lambda$ values, and number of samples.**

| Domain | Model Sizes | | | | $\lambda$ Values | | | Number of Samples | | |
|---|---|---|---|---|---|---|---|---|---|---|
| | 60M | 82M | 124M | 210M | $\lambda = 1$ | $\lambda = 10$ | $\lambda = 100$ | 2k | 4k | 8k |
| Arxiv | 0.084 | 0.083 | 0.087 | 0.093 | 0.080 | 0.083 | 0.087 | 0.079 | 0.083 | 0.081 |
| Book | 0.158 | 0.164 | 0.157 | 0.165 | 0.159 | 0.164 | 0.173 | 0.165 | 0.164 | 0.170 |
| CC | 0.170 | 0.202 | 0.169 | 0.170 | 0.178 | 0.202 | 0.201 | 0.193 | 0.202 | 0.209 |
| C4 | 0.271 | 0.247 | 0.288 | 0.259 | 0.243 | 0.247 | 0.258 | 0.253 | 0.247 | 0.241 |
| Github | 0.073 | 0.082 | 0.072 | 0.089 | 0.097 | 0.082 | 0.057 | 0.079 | 0.082 | 0.066 |
| Stackexchange | 0.152 | 0.149 | 0.153 | 0.145 | 0.152 | 0.149 | 0.128 | 0.138 | 0.149 | 0.143 |
| Wikipedia | 0.072 | 0.073 | 0.074 | 0.078 | 0.081 | 0.073 | 0.095 | 0.093 | 0.073 | 0.090 |

### B.6. Additional baseline: Data Mixing Laws

We add an additional baseline, Data Mixing Laws (Ye et al., 2024), for comparison. It derives domain weights by leveraging scaling laws of training steps, model sizes, and data mixtures to predict the performance of large models trained on diverse data from small-scale training. This requires training multiple small proxy models with varying domain weights, making it more computationally expensive than ours, which trains just one proxy model.

We use their reported domain weights to train a 684M model on Slimpajama. Since their weights are optimized with the Pile as the target, they may be suboptimal for SlimPajama. However, given the alignment of their objectives and overlap in data sources, we consider the comparison meaningful.

As shown in Table 17 and Table 18, CHAMELEON outperforms Data Mixing Laws in both perplexity and downstream tasks

at a fraction of the cost. Data Mixing Laws' FLOPs is calculated for 4 different proxy sizes and 20 separate mixtures, where our cost is 2 orders of magnitude lower.

Table 17: **Data Mixing Laws - perplexity**. Perplexity in the same settings as Table 2.

| Domain | CHAMELEON | Data Mixing Laws |
|---|---|---|
| Arxiv | 8.31 | 7.55 |
| Book | 39.23 | 45.06 |
| CC | 40.11 | 44.21 |
| C4 | 42.59 | 45.79 |
| Github | 4.20 | 4.01 |
| Stackexchange | 7.94 | 7.96 |
| Wikipedia | 13.90 | 16.20 |
| Average PPL ($\downarrow$) | **22.31** | 24.40 |
| # Domains Over Uniform | **4/7** | 4/7 |
| FLOPs | $\mathbf{1.36 \times 10^{17}}$ | $5.36 \times 10^{19}$ |
| | $(1\times)$ | $(394\times)$ |

Table 18: **Data Mixing Laws - reasoning.** Accuracy of downstream tasks in the same settings as Table 2.

| Benchmark | CHAMELEON | Data Mixing Laws |
|---|---|---|
| ARC-E | 37.8 | 34.5 |
| COPA | 61.9 | 59.0 |
| HellaSwag | 27.1 | 27.4 |
| Lambada | 15.1 | 14.7 |
| LogiQA | 22.6 | 26.0 |
| MultiRC | 57.2 | 57.2 |
| OpenBook | 14.4 | 25.2 |
| PiQA | 60.5 | 58.5 |
| QQP | 39.2 | 36.8 |
| RACE | 26.5 | 26.4 |
| SciQ | 64.3 | 57.2 |
| Social IQA | 35.7 | 36.1 |
| WinoGrande | 52.1 | 48.4 |
| Average ($\uparrow$) | **39.6** | 39.0 |

## B.7. Domain weights on Pile

We report the domain weights we use on the Pile dataset in Table 19. Note that CHAMELEON and DoGE are from our own experiments, Human is suggested in (Gao et al., 2020), DoReMi uses the same as (Xie et al., 2023), and RegMix uses weights from (Liu et al., 2024).

## B.8. Domain weights for finetuning

We report the $\alpha^{\mathrm{FT}}$ on Wiki40b and Stack datasets separately below, which corresponds to Section 4.3.

## B.9. PPL of finetuning.

We report the results of fine-tuning with $\alpha^{\mathrm{PT}}$ in Table 22 and Table 23 for reference. It is clear to see that fine-tuning with KRLS-based domain weights is better than the one with the inverse of KRLS-based weights.

Table 19: **Domain weights on Pile.**

| Domain | Human | DoReMi | DoGE | RegMix | CHAMELEON |
|---|---|---|---|---|---|
| Arxiv | 0.134 | 0.004 | 0.0608 | 0.001 | 0.0386 |
| Dm_mathematics | 0.025 | 0.002 | 0.0280 | 0.000 | 0.0538 |
| Enron_emails | 0.004 | 0.009 | 0.0239 | 0.002 | 0.0085 |
| Europarl | 0.005 | 0.008 | 0.0407 | 0.000 | 0.0048 |
| Freelaw | 0.049 | 0.005 | 0.0293 | 0.001 | 0.0147 |
| Github | 0.054 | 0.022 | 0.0693 | 0.000 | 0.0099 |
| Gutenberg_pg_19 | 0.025 | 0.009 | 0.0283 | 0.002 | 0.0115 |
| Hackernews | 0.010 | 0.016 | 0.3949 | 0.012 | 0.0637 |
| Nih_exporter | 0.007 | 0.008 | 0.180 | 0.001 | 0.0424 |
| Philpapers | 0.003 | 0.034 | 0.0266 | 0.000 | 0.0226 |
| Pile_cc | 0.142 | 0.743 | 0.0348 | 0.870 | 0.4519 |
| Pubmed_abstracts | 0.107 | 0.014 | 0.0398 | 0.024 | 0.0104 |
| Pubmed_central | 0.136 | 0.006 | 0.0251 | 0.003 | 0.1207 |
| Stackexchange | 0.118 | 0.019 | 0.0266 | 0.000 | 0.0226 |
| Ubuntu_irc | 0.009 | 0.011 | 0.0474 | 0.064 | 0.0123 |
| Uspto_backgrounds | 0.053 | 0.004 | 0.0366 | 0.002 | 0.0212 |
| Wikipedia_en | 0.117 | 0.086 | 0.0425 | 0.016 | 0.1075 |

Table 20: **Wiki40b Domain Weights.**

| Domain | CHAMELEON |
|---|---|
| French | 0.115 |
| German | 0.163 |
| Italian | 0.127 |
| Spanish | 0.109 |
| Portuguese | 0.090 |
| Dutch | 0.140 |
| Polish | 0.257 |

Table 21: **Stack Dataset Training Weights.**

| Domain | CHAMELEON |
|---|---|
| Python | 0.125 |
| Java | 0.129 |
| C | 0.102 |
| C++ | 0.088 |
| Go | 0.241 |
| Ruby | 0.118 |
| PHP | 0.197 |

Table 22: **Per-domain perplexity compared with the Uniform baseline for fine-tuning with 684M parameter models on the 7 languages in the Wiki40b dataset.**

| Domain | Uniform | CHAMELEON ($\alpha^{\text{FT}}$) | CHAMELEON ($\alpha^{\text{PT}}$) |
|---|---|---|---|
| French | 6.86 | 6.51 | 7.14 |
| German | 10.12 | 8.78 | 10.85 |
| Italian | 13.29 | 12.42 | 14.42 |
| Spanish | 8.41 | 8.04 | 8.70 |
| Portuguese | 8.00 | 7.78 | 8.14 |
| Dutch | 13.98 | 12.30 | 15.05 |
| Polish | 5.07 | 4.21 | 5.31 |
| Average PPL ($\downarrow$) | 9.43 | 8.58 | 9.94 |

Table 23: **Per-domain perplexity compared with the Uniform baseline for fine-tuning with 684M parameter models on the 7 languages in the Stack dataset.**

| Domain | Uniform | CHAMELEON ($\alpha^{\text{FT}}$) | CHAMELEON ($\alpha^{\text{PT}}$) |
|---|---|---|---|
| Python | 19.98 | 16.53 | 20.11 |
| Java | 19.27 | 15.53 | 19.32 |
| C | 28.24 | 22.58 | 25.02 |
| C++ | 25.16 | 21.09 | 23.83 |
| Go | 30.25 | 19.26 | 28.66 |
| Ruby | 21.78 | 17.83 | 21.75 |
| PHP | 9.45 | 7.43 | 9.47 |
| Average PPL ($\downarrow$) | 22.02 | 17.18 | 21.17 |

