# OpenReview forum: "Chameleon: A Flexible Data-mixing Framework for Language Model Pretraining and Finetuning"
_ICML.cc/2025/Conference — ICML 2025 poster_

### Official Review · Reviewer_jzfM · 2025-03-11

**Overall Recommendation:** 3

**Summary:**

This manuscript introduces a flexible data-mixing framework for LLM that uses kernel ridge leverage scores (KRLS) computed from learned domain embeddings using a proxy model. It quantifies each domain’s representativeness and interdependence within the embedding space, and then uses this information to generate a weighted data mixture for both pretraining and finetuning.  The comprehensive experiments and ablation studies demonstrate its practical benefits on both perplexity and downstream tasks by outperforming existing methods such as DoReMi and DoGE with a small computational overhead.

## Post Rebuttal ##

Most of my concerns are properly addressed. I would like to raise my score.

## Post Rebuttal ##

**Claims And Evidence:**

Overall, most claims are supported by empirical evidence. However, some of the claims are not supported by clear and convincing evidence.

1. The authors claim that the method is computationally efficient. However, the submission does not include a detailed runtime analysis or complexity comparison with baseline methods.

2. The novelty of the manuscript is the ability to quantify each domain’s representativeness and interdependence in the learned embedding. However, the evidence provided does not clearly demonstrate how reliable or robust these kernel ridge leverage scores are across different scenarios. More analysis or visualization of these scores and their correlation with the performance would strengthen this claim.

3. The authors claim that the computed domain weights transfer directly to new data and different model scales without needing to retrain the proxy model. Although the provided experimental results are promising, the evidence could be more convincing if more experiments or detailed ablation studies were provided to thoroughly evaluate this transferability under a wider range of conditions.

4. The paper would be strengthened by an in-depth theoretical analysis explaining why kernel ridge leverage scores are particularly effective for domain reweighting.

**Essential References Not Discussed:**

N/A

**Experimental Designs Or Analyses:**

Yes, the experimental design and analyses are sound. however, I would suggest the author to include a runtime analysis to validate the claim that the proposed method is computationally efficient.

**Methods And Evaluation Criteria:**

The proposed methods and evaluation criteria align well with the problem of efficient data mixing for LLM training. The use of kernel ridge leverage scores computed from learned domain embeddings makes sense for quantifying domain representativeness and interdependence. Measuring improvements in perplexity and downstream task performance are standard metrics for evaluating language model training.

**Other Comments Or Suggestions:**

Please see the Weaknesses section.

**Other Strengths And Weaknesses:**

#### Strengths:

1. The paper presents a novel method that leverages the kernel ridge leverage score to reweight data domains, which is a combination of ideas from data mixing, transfer learning, and efficient training of LLMs.

2. The proposed work holds significance in practice as it is importance in scaling LLM training without retraining expensive proxy models.

3. The extensive empirical results indicate improvements in both pretraining and finetuning, which validates the practical values of the proposed method.

4.  writing is clear and observation-driven, which makes it easy for readers to follow along and engage with the authors to approach the problem.

#### Weaknesses:

1. While the paper claims the computational efficiency, it lacks a detailed runtime analysis or formal complexity comparison with baseline methods.

2. The theoretical justification behind using KRLS is largely supported by empirical evidence rather than formal proofs.

3. The clarity regarding the transferability of domain weights across diverse datasets and model sizes could be improved with more detailed ablation studies and analysis.

**Questions For Authors:**

1. Could you provide a more detailed runtime analysis and complexity comparison of your method with baseline methods?

2. Can you elaborate on the theoretical insights behinds why KRLS are effective for domain reweighting?

3. Could you discuss in more detail how the computed domain weights transfer across different datasets and model scales?

**Relation To Broader Scientific Literature:**

The key contributions of the paper are well situated within the broader scientific literature, building upon and extending several established ideas, which include (1) kernel methods and leverage scores; (2) data selection and mixing in LLM Training; and (3) domain adaptation and representation kearning.

**Theoretical Claims:**

I have checked the theoretical proof in the appendix. The proof of the equivalence between the ridge leverage scores computed in the feature space and those obtained via kernel ridge regression when using a linear kernel. The proof, presented as Lemma A.1, correctly applies standard matrix identities to show that the diagonal elements of the corresponding hat matrices are equal. While some intermediate steps are missing, I did not find any  logic issues.

Beyond this lemma, most theoretical claims are supported by empirical evidence and ablation studies rather than fully formalized proofs. Overall, the proofs that were provided are correct, and no major issues were found.

---

> ### Author Rebuttal · Authors · 2025-03-31
>
> We thank the reviewer for their valuable feedback and address all remaining concerns below:
>
> > Q1. Runtime analysis and complexity comparison
>
> Obtaining embeddings $x_i, \, i=1,\ldots,k$ requires a single forward pass for each $a \in B_i$ through the proxy $h_{\theta_p}(a)$; inference is fast as the proxy is a small model. Computing (KRLS) involves inverting a matrix of size $k \times k$ in $\mathcal{O}(k^3)$, which is computationally cheap since datasets typically have a small number $k$ of domains. We do not add any overhead in proxy training. In contrast, DoGE requires per-domain gradient computation at each iteration, and DoReMi runs inference of the reference model for perplexity comparisons.
>
> For a straightforward comparison, we report GPU hours below. **DoReMi and DoGE incur over 10% of the base model training cost, while Chameleon reduces it to under 2%.** These savings are particularly impactful for academic labs with limited computational resources.
>
> **Table jzfM-1**: Runtime comparison.
> |Method|GPU Hours|
> |-|:-:|
> |DoReMi|7.4|
> |DoGE|6.3|
> |Chameleon|0.8|
> |684M base model|56|
>
> See more details on the FLOPs computations in *Q2 of Reviewer ubpq*.
>
> > Q2. Theoretical insights for the effectiveness of KRLS in domain reweighting
>
> We use kernel ridge leverage scores (KRLS) to determine domain weights. KRLS is a well-established tool in data analysis. It quantifies the influence or importance of data points [Alaoui & Mahoney, 2015]. This property is leveraged in machine learning for tasks like density estimation [Pauwels et al., 2018] and novelty detection [Ducharlet et al., 2024; Lasserre & Pauwels, 2019].
>
> The inverse KRLS is proportional to the Christoffel function value [Pauwels et al., 2018]. This relationship provides additional theoretical justification for our approach. Christoffel functions (Eq. (1) in [Pauwels et al., 2018]) precisely characterize the local density of the data distribution in the feature space, where higher values indicate denser regions.
>
> We compute the score $S_\lambda(D_i)$ of domain $i$ using Eq. (KRLS) on page 4. During pretraining, assigning higher sampling probability to domains with low KRLS (and thus high $S_\lambda^{-1}$/Christoffel value) upweights high-density data regions, which are most influential on base LMs' performance [1]. LLM finetuning aims to specialize on a novel specific task, requiring the model to learn differential features not fully captured during pretraining, so we instead prioritize the domains with high $S_\lambda$. Section 3.2 converts either $S_\lambda^{-1}$ or $S_\lambda$ into probability distributions $\alpha$ by appliying softmax normalization.
>
> We will revise Section 3.2 to explicitly link the data mixing goal to KRLS and inverse KRLS, grounding it in statistical learning theory.  We will make this discussion self-contained within the main text, incorporating analysis from Appendix A.
>
> > Q3. More detail on how domain weights transfer across datasets and model scales
>
> - **Transfer across model sizes**: Prior works (e.g., DoReMi, DoGE, RegMix) have shown that domain weights transfer well across model scales. To further validate this, we trained 1.2B models on SlimPajama and found that weights from an 82M proxy model effectively transfer to both 684M and 1.2B models. Notably, Chameleon achieves even greater improvements on larger models, highlighting its scalability.
>
> **Table jzfM-1:** PPL with 1.2B model
> |Domain|Uniform|DoReMi|DoGE|Chameleon|RegMix|
> |-|:-:|:-:|:-:|:-:|:-:|
> |Arxiv|6.30|7.09|7.07|6.33|10.61|
> |Book|28.25|32.66|27.83|24.63|27.55|
> |CC|31.19|29.96|28.11|26.95|24.70|
> |C4|34.74|33.05|31.06|29.58|31.94|
> |Github|2.91|3.03|3.07|2.94|4.08|
> |Stackexchange|6.01|6.44|5.80|5.76|9.54|
> |Wikipedia|8.65|7.93|10.88|9.03|20.08|
> |*Average PPL*|16.86|17.17|16.26|**15.03**|18.36|
>
> **Table jzfM-2:** Downstream accuracy with 1.2B model
> |Task|Uniform|DoReMi|DoGE|Chameleon|RegMix|
> |-|:-:|:-:|:-:|:-:|:-:|
> |ARC-E|39.4|41.2|41.9|42.4|43.0|
> |COPA|64.0|66.0|63.0|61.0|66.0|
> |HellaSwag|27.5|27.7|28.2|28.4|27.6|
> |Lambada|17.9|17.3|18.7|21.6|20.7|
> |LogiQA|22.0|24.0|22.0|21.2|20.7|
> |MultiRC|57.2|57.2|57.2|57.2|56.9|
> |OpenBookQA|15.0|13.6|13.8|16.4|17.4|
> |PiQA|61.5|61.9|61.8|63.8|58.7|
> |QQP|36.8|36.8|36.9|36.9|36.8|
> |RACE|26.0|26.7|27.8|29.1|28.4|
> |SciQ|69.7|68.3|69.0|72.6|72.0|
> |SocialIQA|36.2|36.5|35.9|37.2|36.1|
> |WinoGrande|52.8|49.6|48.9|51.5|50.0|
> |*Average*|40.5|40.5|40.4|**41.5**|41.1|
>
> - **Transfer across datasets**: We conduct ablation studies on the Pile (Section 4.2). Specifically, we retrain a proxy model on the Pile for reference. As shown in [Domain weights on the Pile](https://imgur.com/a/2Xnk56t), the weight from a proxy trained on the Pile (the blue column) aligns with the weights transferred from proxies trained on SlimPajama at various sizes (the other columns), confirming Chameleon’s robust transferability.
>
> [1] Mallen et al. When not to trust language models: Investigating effectiveness of parametric and non-parametric memories. ACL (2023).

---

> > ### Comment · Reviewer_jzfM · 2025-04-07
> >
> > Thank you for the response. Most of my concerns are properly addressed.

---

### Official Review · Reviewer_ubpq · 2025-03-11

**Overall Recommendation:** 3

**Summary:**

This paper introduces a new data mixing framework for language pretraining and finetuning wherein the mixing weights for different domains are constructed from a domain affinity matrix generated via kernel functions on domain embeddings. This domain matrix can naturally be transformed into domain weights for pre-training (i.e., emphasizing broader and different domains) and finetuning (emphasizing similar domans). The key advantage of this framework is that it does not rely extensively on training proxy models, and thereby can be seen as a low-cost alternative to existing data mixing frameworks.

**Claims And Evidence:**

The paper is well-supported by rigorous numerical analysis.

**Essential References Not Discussed:**

The paper comprehensively covers the primary literature.

**Experimental Designs Or Analyses:**

The experiments are meaningful and sound.

**Methods And Evaluation Criteria:**

Yes

**Other Comments Or Suggestions:**

N/A

**Other Strengths And Weaknesses:**

Strengths:
- The paper is well-written and timely, addressing rigorous experiments


Weaknesses:
- The reasoning behind obtaining domain weights from the KRLS is unclear. Is there some theoretical relationship or connection that can be derived to show why pretraining weights as designed, or finetuning weights as designed, are appropriate? I appreciate that the authors have provided some intuition but it would be beneficial to get more insight.
- There doesn't seem to be a significant performance improvement from the mixing law, and indeed, the major selling point is that it achieves a competitive performance at orders of magnitude lower cost. It would be useful to emphasize this point further, especially in numerical results to better break down all cost calculations (e.g., in domain transfer).
- It would be useful to include Data Mixing Laws as a baseline, at least for experiments on generalization [1]

[1] Ye, Jiasheng, et al. "Data mixing laws: Optimizing data mixtures by predicting language modeling performance." arXiv preprint arXiv:2403.16952 (2024).

**Questions For Authors:**

Refer to Strengths & Weaknesses

**Relation To Broader Scientific Literature:**

The paper adds to the data mixing literature, with specific focus on compute-efficiency and adaptation to new tasks.

**Theoretical Claims:**

There are some theoretical results in the Appendix, which I briefly reviewed.

---

> ### Author Rebuttal · Authors · 2025-03-31
>
> We thank the reviewer for their valuable feedback and address all remaining concerns below:
>
> > Q1. Reasoning behind obtaining domain weights from the KRLS
>
> We use kernel ridge leverage scores (KRLS) to determine domain weights. KRLS is a well-established tool in data analysis. It quantifies the influence or importance of data points [Alaoui & Mahoney, 2015]. This property is leveraged in machine learning for tasks like density estimation [Pauwels et al., 2018] and novelty detection [Ducharlet et al., 2024; Lasserre & Pauwels, 2019].
>
> The inverse KRLS is proportional to the Christoffel function value [Pauwels et al., 2018]. This relationship provides additional theoretical justification for our approach. Christoffel functions (Eq. (1) in [Pauwels et al., 2018]) precisely characterize the local density of the data distribution in the feature space, where higher values indicate denser regions.
>
> We compute the score $S_\lambda(D_i)$ of domain $i$ using Eq. (KRLS) on page 4. During pretraining, assigning higher sampling probability to domains with low KRLS (and thus high $S_\lambda^{-1}$/Christoffel value) upweights high-density data regions, which are most influential on base LMs' performance [1]. LLM finetuning aims to specialize on a novel specific task, requiring the model to learn differential features not fully captured during pretraining, so we instead prioritize the domains with high $S_\lambda$. Section 3.2 converts either $S_\lambda^{-1}$ or $S_\lambda$ into probability distributions $\alpha$ by appliying softmax normalization.
>
> We will revise Section 3.2 to explicitly link the data mixing goal to KRLS and inverse KRLS, grounding it in statistical learning theory.  We will make this discussion self-contained within the main text, incorporating analysis from Appendix A.
>
> > Q2. Computational cost breakdown
>
> Chameleon's main computational cost comes from 1) proxy training and 2) embedding extraction, with proxy training being dominant. In our setting, training an 82M proxy model requires $10^{17}$–$10^{18}$ FLOPs, while DoReMi and DoGE take longer to converge, leading to 5-10x higher costs (see line 299, "Stability and Practicality"). Regarding our embedding extraction, it requires only $10^{15}$ FLOPs (<1% of proxy training). Importantly, Chameleon avoids proxy retraining when domains change, incurring only embedding extraction costs. In contrast, DoReMi and DoGE induce their proxy retraining FLOPs.
>
> Our method is also significantly cheaper in GPU hours, see *Response to Q1 of Reviewer jzfM* for more details and complexity analysis.
>
> Beyond efficiency, Chameleon is also more stable, making it resource-efficient in practical use. Unlike DoReMi and DoGE, which are sensitive to hyperparameters, Chameleon remains robust, see *Q2 of Reviewer jX7F* for more details.
>
> Lastly, we note that Chameleon shows favorable accuracy behaviour on larger models as well, as shown in our additional 1.2B model experiments (see *Q3 of Reviewer jzfM*).
>
> > Q3. Data Mixing Laws
>
> We first provide discussions and then present empirical comparisons.
>
> Data Mixing Laws derive domain weights by leveraging scaling laws of training steps, model sizes, and data mixtures to predict the performance of large models trained on diverse data from small-scale training. This requires training multiple small proxy models with varying domain weights, making it more computationally expensive than ours, which trains just one proxy model.
>
> We use their reported domain weights to train a 684M model on Slimpajama. Since their weights are optimized with the Pile as the target, they may be suboptimal for SlimPajama. However, given the alignment of their objectives and overlap in data sources, we consider the comparison meaningful.
>
> **Chameleon outperforms Data Mixing Laws in both perplexity and downstream tasks** at a fraction of the cost. Data Mixing Laws' FLOPS is calculated for 4 different proxy sizes and 20 separate mixtures, where **our cost is 2 orders of magnitude lower**.
>
> **Table ubpq-1:** PPL comparison with Data Mixing Laws
> ||Data Mixing Laws|Chameleon|
> |-|:-:|:-:|
> |Arxiv|7.55|8.31|
> |Book|45.06|39.23|
> |CC|44.21|40.11|
> |C4|45.79|42.59|
> |Github|4.01|4.20|
> |Stackexchange|7.96|7.94|
> |Wikipedia|16.20|13.90|
> |Avg PPL|24.40|**22.31**|
> |# Domains Over Uniform|4/7|4/7|
> |FLOPs|$5.36×10^{19}$|$1.36×10^{17}$|
>
> **Table ubpq-2:** Downstream accuracy comparison with Data Mixing Laws
> |Task|Data Mixing Laws|Chameleon|
> |-|:-:|:-:|
> |ARC-E|34.5|37.8|
> |COPA|59.0|61.9|
> |HellaSwag|27.4|27.0|
> |Lambada|14.7|15.1|
> |LogiQA|26.0|22.6|
> |MultiRC|57.2|57.2|
> |OpenBook|25.2|14.4|
> |PiQA|58.5|60.5|
> |QQP|36.8|39.2|
> |RACE|26.4|26.5|
> |SciQ|57.2|64.3|
> |Social IQA|36.1|35.7|
> |WinoGrande|48.4|52.1|
> |Average|39.0|**39.6**|
>
> [1] Mallen et al. When not to trust language models: Investigating effectiveness of parametric and non-parametric memories. ACL (2023).
>
> [2] Parmar et al. Data, data everywhere: A guide for pretraining dataset construction. ACL 2024.

---

### Official Review · Reviewer_jX7F · 2025-03-19

**Overall Recommendation:** 3

**Summary:**

Authors propose a method for data sampling for pretraining and finetuning language models. Their idea is to train a classifier, then extract the middle layer word embeddings of the classifier for each domain in the training data, and then to do matrix factorization to obtain a scalar weight for each domain. These weights are used in two heuristic equations to obtain the probabilities for sampling from the domains during the training of the LM.

It is empirically shown that the algorithm is faster than the baselines, and can be used without retraining the classifier when new data is added to the training. It is also shown that the algorithm can be used during finetuning.

=================

UPDATE: I updated my review score.

**Claims And Evidence:**

The claims on speed need further explanations, see the section below

**Essential References Not Discussed:**

N/A

**Experimental Designs Or Analyses:**

Most of them

**Methods And Evaluation Criteria:**

Yes

**Other Comments Or Suggestions:**

See above

**Other Strengths And Weaknesses:**

**Strengths:**

-  The method is simple, intuitive, and easy to implement.
-  The paper is well written, and the experiments are well organized.
-  The topic is very relevant and timely.

**Weaknesses:**

- The core idea is just a heuristic (in authors words): pretaining needs "broadly shared semantic structures" and finetuning needs "distinct and unique data" to "highlight domain-specific characteristics".
  To me the statements above are just some vague justifications for what empirically works.
- In my opinion the performance improvements are virtually non existent--compared to the model DoGE. The main distinction lies in the speed. Authors might argue that when new data is added, the performance improvement is more tangible. But if we put speed aside, and retrain the baseline proxy networks, then again the only distinction becomes speed. In general the proxy model is relatively small, and its training data is a fraction of the entire training data. How many GPU hours are needed to train the proxy networks across the models? Does increasing speed in training this network make any significant difference in energy consumption? How often "retraining" the proxy network is needed at all? Is it needed at all?

 I would be happy to revise my score if authors give me convincing answers.

**Other comments:**

Please don't force the reviewers to read your appendix.

**Questions For Authors:**

See above

**Relation To Broader Scientific Literature:**

Builds upon existing literature, primarily DoGE

**Theoretical Claims:**

Some of them

---

> ### Author Rebuttal · Authors · 2025-03-31
>
> We thank the reviewer for their valuable feedback and address all remaining concerns below:
>
> > Q1. Theoretical motivation
>
> We use kernel ridge leverage scores (KRLS) to determine domain weights. KRLS is a well-established tool in data analysis. It quantifies the influence or importance of data points [Alaoui & Mahoney, 2015]. This property is leveraged in machine learning for tasks like density estimation [Pauwels et al., 2018] and novelty detection [Ducharlet et al., 2024; Lasserre & Pauwels, 2019].
>
> The inverse KRLS is proportional to the Christoffel function value [Pauwels et al., 2018]. This relationship provides additional theoretical justification for our approach. Christoffel functions (Eq. (1) in [Pauwels et al., 2018]) precisely characterize the local density of the data distribution in the feature space, where higher values indicate denser regions.
>
> In our context, we compute the score $S_\lambda(D_i)$ of domain $i$ using Eq. (KRLS) on page 4. During pretraining, assigning higher sampling probability to domains with low KRLS (and thus high $S_\lambda^{-1}$/Christoffel value) upweights high-density data regions, which are most influential on base LMs' performance [1]. LLM finetuning aims to specialize on a novel specific task, requiring the model to learn differential features not fully captured during pretraining, so we instead prioritize the domains with high $S_\lambda$. Section 3.2 converts either $S_\lambda^{-1}$ or $S_\lambda$ into probability distributions $\alpha$ by appliying softmax normalization.
>
> We will revise the phrasings in Section 3.2 to explicitly connect the data mixing goal to the mathematical properties of KRLS and inverse KRLS, clarifying its foundation in theoretical principles from statistical learning rather than just empirical heuristics. We will make this discussion self-contained within the main text, drawing upon the analysis currently in Appendix A.
>
> > Q2. Impact of our computational efficiency
>
> The computational cost associated with determining the domain mixture via proxy training is non-negligible. Table jX7F-1 below reports the required GPU (H100) hours for our experiments in Tab. 2 in the paper. Compared to DoReMi and DoGE, which add over 10% to base model training costs, **we reduce computational overhead to less than 2% of final training cost.** This reduction is crucial for academic labs and smaller-scale training.
>
> **Table jX7F-1:** GPU hours for universal generalization experiments.
> |Method|GPU Hours|
> |-|-|
> |DoReMi|7.4h|
> |DoGE|6.3h|
> |Chameleon|0.8h|
> |684M base model|56h|
>
> Even for larger base models, the computational cost reported is often an optimistic lower bound for the baselines since *DoReMi and DoGE require extensive hyperparameter tuning*. It has been shown that DoReMi's weights are unstable or difficult to reproduce [2; Fan et al., 2024b] and DoGE approximations make it more sensitive to learning rate [Kang et al., 2024b]. We also noticed that DoGE is also extremely sensitive to their Bregman coefficient $\mu$, as shown in Table jX7F-2, where we report domain weights and validation PPL in the last line. Small variations in $\mu$ drastically change domain weights and degrade validation PPL, necessitating repeated validation on base models. This sensitivity contradicts the goal of data mixing methods: weights should transfer reliably to large models without costly grid searches.
>
> **Table jX7F-2:** DoGE's weights are highly sensitive to $\mu$.
> ||$\mathbf{\mu=0.05}$|$\mu=0.01$|$\mu=0.1$|
> |:-:|:-:|:-:|:-:|
> |Arxiv|0.041|0.210|0.222|
> |Book|0.078|0.025|0.069|
> |CC|0.268|0.052|0.068|
> |C4|0.283|0.025|0.050|
> |Github|0.059|0.021|0.378|
> |Stackexchange|0.230|0.649|0.103|
> |Wikipedia|0.041|0.019|0.110|
> |*Avg PPL* of 124M model|24.97|25.45|26.73|
>
> In contrast, Chameleon is stable across training steps, model sizes, $\lambda$, and sample counts (Tables 10, 11). This means **our method can produce promising domain weights without repeated validation**, significantly reducing overall costs for users.
>
> Another key aspect, as the reviewer pointed out, is the cost of incorporating new data sources. Our data-centric approach requires only inference to obtain new embeddings and recompute KRLS, whereas proxy optimization-based methods like DoReMi and DoGE necessitate full retraining and additional tuning.
>
> Lastly, we further validate performance improvement by training 1.2B models. Chameleon demonstrates gains in both perplexity and downstream task accuracy (see *Q3 for Reviewer jzfM*).
>
> [1] Mallen et al. When not to trust language models: Investigating effectiveness of parametric and non-parametric memories. ACL (2023).
>
> [2] Parmar et al. Data, data everywhere: A guide for pretraining dataset construction. ACL 2024.

---

### Decision · Program_Chairs · 2025-05-01

**Decision:**

Accept (poster)

**Comment:**

The manuscript introduces a data-mixing framework leveraging kernel-based scores for data curation in both pretraining and finetuning. The proposed method demonstrates promising empirical results, often at a significantly reduced computational cost compared to existing methods like DoReMi and DoGE.

The review process identified several areas that the authors should address to further strengthen the manuscript for its camera-ready version. A primary concern across multiple reviewers was the need for a more detailed exposition of a principled understanding underpinning the approach, specifically regarding why KRLS is effective for domain reweighting and the intuitive connection between KRLS scores and data mixing strategies. I will not detail additional dimensions for improvement (e.g., better visualizations of KRLS scores or various ablations), but overall incorporating reviewers’ feedback will strengthen the grounding and empirical validity of the work and further clarify the advantages of the proposed method.